# Influences of familiarity and recollection on value-based decision-making

Avinash Rao Vaidya[1,2]*, Johanny Castillo[1], Alejandro Torres[1], David Badre[1,3]

1 Department of Cognitive and Psychological Sciences, Brown University, Providence, Rhode Island, 2 National Institute on Drug Abuse Intramural Research Program, Baltimore, Maryland, 3 Carney Institute for Brain Sciences, Brown University, Providence, Rhode Island

* avinash.vaidya@nih.gov

## Abstract

We regularly retrieve information from memory to inform decisions in daily life. For example, when choosing a place to eat, we may be enticed by a brand name because of its familiarity or drawn to an independent restaurant because of recollections of a delicious lunch we had there once before. Despite the centrality of memory in such everyday choices, it remains unclear how these different memory processes (i.e., familiarity versus recollection) interact during value judgment and decision-making. Here we describe a novel experimental paradigm that tests the contributions of these processes to risk-based choice. In this task, participants had to retrieve the source of an image from an earlier encoding task to infer the probability of a bet being rewarded. Some images were repeated multiple times at encoding, while others only appeared once and others were lures that never appeared during the encoding task. We examined behavior in this task across two experiments, one conducted fully online and the second both online and in-laboratory. We found that subjective value increased with familiarity during memory-based decision-making. Betting on lure items even increased with false familiarity. Further, we observed evidence that familiarity and source value information interacted, such that the relationship of both familiarity and source value information with betting were increased when both were high. Our results highlight the importance of subjective familiarity in decision-making and potentially indirectly increasing the value of information retrieved from source memory.

## Introduction

Memory informs how we construct preferences and evaluate options during our daily decision-making [1]. For example, when choosing where to go for dinner, we might recall the uniquely enjoyable meal we had at a restaurant, or alternatively, the case of indigestion that followed the next day. Many studies have demonstrated that items associated with higher values are preferentially encoded into memory [2–5], and that

**Data availability statement:** Data and analysis code for the experiments described here are available at the Memory Betting Task repository: https://osf.io/35p4r/?view_only=5f-64c753ebfd48acb7f3cb04fa7c7c97. A DOI will be provided after this repository is made public upon acceptance.

**Funding:** DB Office of Naval Research MURI-N00014-16-1-2832 https://www.onr.navy.mil/ No (Award supports first author) ZIA DA000642 National Institutes on Drug Abuse https://nida.nih.gov/ No The funders had no role in study design, data collection and analysis, decision to publish, or preparation of the manuscript.

**Competing interests:** The authors have declared that no competing interests exist.

value-relevant information from specific events is also recalled to assist decision-making [6–8]. Likewise, patients with medial temporal lobe damage make more stochastic preference-based choices [9], suggesting that retrieval of information from episodic memory forms an important component of value judgment.

The strength of encoding and retrieval of information about options can also influence their perceived subjective value. Several studies have shown that options for which values and identity can be recollected are preferred in choice tasks, even when the values of these options are no better than the average value of alternatives [10–13]. Other work has shown that the value attributed to items forgotten from a word list is reduced, relative to those that can be recalled [14,15]. Thus, retrieval does not just inform value judgment through access to details that inform the decision, but the quality of memory processing itself might shape the value of options. Another way of interpreting these findings is that items that are better encoded or are more easily retrieved are assumed to have greater utility [16], and so are attributed greater expected value.

A parallel line of work has focused on how familiarity can also directly influence value judgment. Familiarity refers to the sense of having previously observed an item absent any specific spatiotemporal context [17]. Mere exposure to a novel, affectively neutral stimulus, is enough to shift preferences towards greater liking on a rating scale [18–20], even under conditions when participants do not report observing the stimulus [21,22]. Familiarity also affects behavior during risky decision-making: participants are averse to betting on novel stimuli and are less willing to bet on relatively unfamiliar stimuli more generally [23].

Familiarity and recollection are often described as distinct processes that rely on dissociable neural substrates [17,24], though defining behavioral measures that distinguish between these processes has proven to be challenging [25,26]. From the perspective of a rational economic agent, a non-specific familiarity signal should not increase expected value as it does not carry information about previous positive or negative experience with an option, or the outcomes of previous interactions with that option. For such an agent, recollection of value-relevant information about an option should be given priority in decision-making, as this information reduces ambiguity about the value of an option. However, if previous experiences are assumed to be neutral, it is possible that a familiar option might be perceived as safe compared to the unknown value of novel alternatives [20]. To our knowledge, no studies to date have examined the interaction of familiarity and recollection in the context of risky decision making, making it unclear how these processes interact to impact choice.

To test the contributions of recollection and familiarity in the same setting, we developed a memory-based decision task where the values of options were not related to the strength of encoding. Participants were asked to recollect trial-unique source information about a prior encounter with a stimulus to determine the probability of a bet being rewarded. Importantly, these stimuli had been encoded with different numbers of repetitions in an earlier phase and so varied in their familiarity and the strength of encoding. Unlike other past research in this domain, information about the value of these stimuli was not available at encoding. Value information was only

signaled indirectly through source associations formed at encoding, but unknown to the participant until the betting stage. This design allowed us to decouple strength of encoding from option values.

We set out to test the following four hypotheses using this task:

1.  Given the known effects of familiarity on value judgment, we expected that greater familiarity should increase the expected value of options – even for options with higher levels of risk.

2.  Assuming a rational economic perspective, familiarity should only increase the values of options in the absence of recollection. That is, we predicted that this non-specific signal should be discounted when more directly relevant value information is retrieved.

3.  Given that retrieval also shapes option values and choice [10,14], we further expected that recollection and confidence in recollected information would increase the value of options independent of the information retrieved.

4.  Retrieval of value-relevant information should drive more accurate decision-making (i.e., better use of the real value of these options).

## Methods: Experiment 1

### Incidental encoding task

Participants first completed an encoding task where they saw 256 photos of scenes [27] in four quadrants of a 2 x 2 grid and were asked to complete an incidental encoding task by classifying these images as indoors or outdoors (Fig 1a). The strength of this encoding varied with the number of stimulus presentations, with half of these scenes presented just once, and half appearing three times during this phase of the experiment. Each scene appeared in one of the four quadrants, with stimuli presented multiple times always appearing in the same quadrant on each presentation. The spacing of repeated presentations was randomized over the course of this encoding task, so there was no correlation between scenes in the one versus three presentation condition appearing earlier or later in the encoding task. Participants classified images by pressing the 'q' or 'p' keys on their keyboard, corresponding to text prompts on the left and right side of the screen indicating whether the scene was indoors or outdoors. The mapping between the keys and the indoor/outdoor classification was randomized trial-to-trial. Participants completed this encoding task in four blocks, each consisting of 128 trials. Each scene image appeared on-screen for three seconds. After participants responded, the text prompt corresponding to their response was underlined until the trial terminated. Trials were separated by a jittered inter-trial interval generated from a lognormal distribution with a mean of 1.0 s and standard deviation of 1.2 s.

### Practice betting task

All participants completed a short practice (20 trials) of the risk-based betting task. The purpose of this practice task was to familiarize participants with the cues and probabilities for risky-decision making in the main task, and to act as a distraction during the delay separating the main memory-based betting task from the encoding task (Fig 1b). In this task, participants learned that four color swatches were associated with different risk levels for rewarding a bet (red: 10%, light-red: 40%, light-green: 60%, green: 90%). A reference for these reward probabilities and the color swatches was displayed throughout this task in the same 2 x 2 grid used in the encoding task, though there was no mention made of any relation between the two grids. The positions of these colors on this grid were randomized for each participant so that each quadrant was equally likely of being associated with higher or lower risk across participants.

On each trial, a square swatch in one of these colors was presented in the center of the screen (below the reference grid), and participants were offered the opportunity to either bet on earning a bonus of $5 or earning nothing at the risk-level indicated by the swatch, or choosing to pass and accepting the safe offer of $2. Participants chose to bet or pass using the 'q' and 'p' keys on a keyboard, with the text prompts for each option randomly presented on either the left or

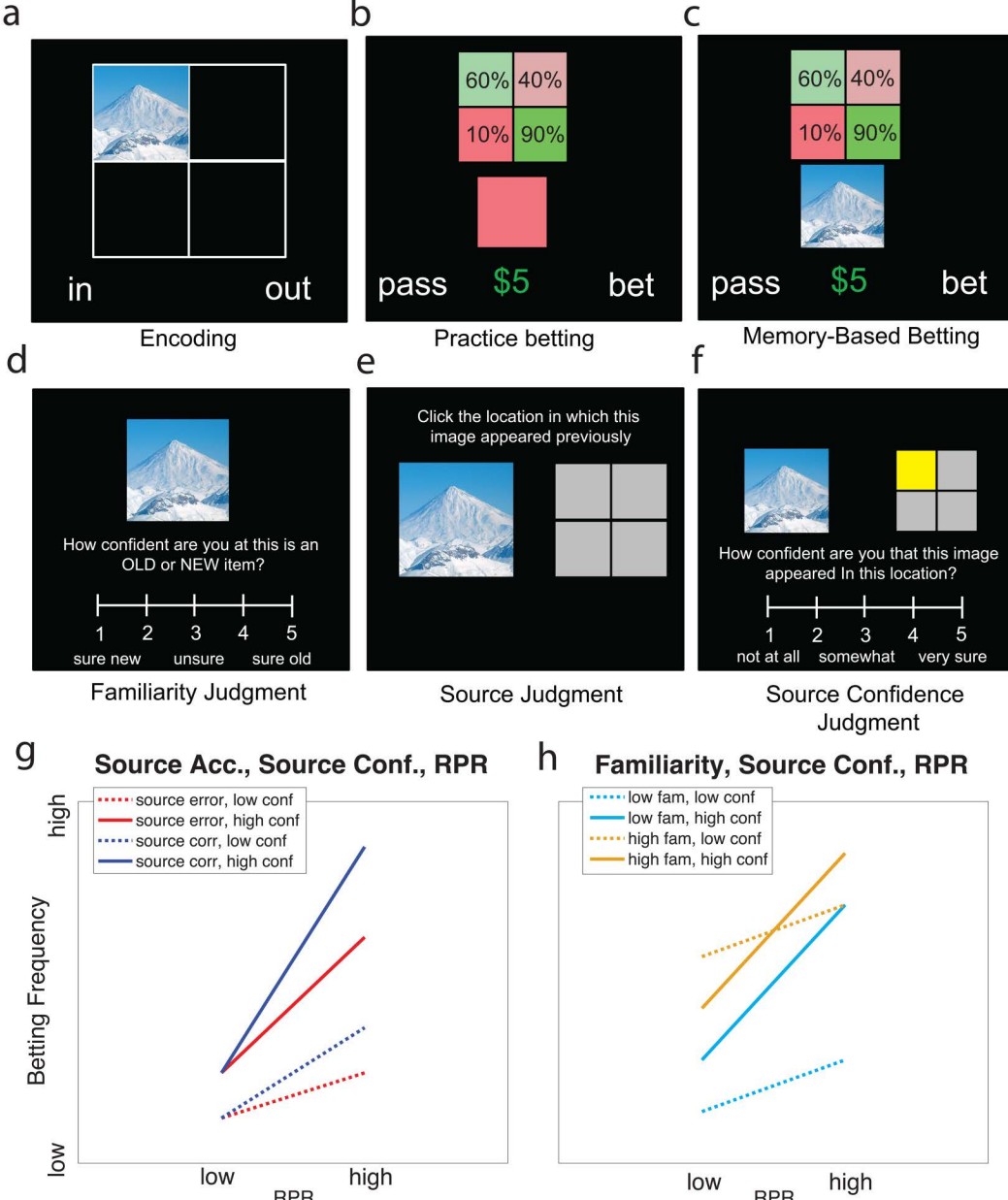

**Fig 1. Schematic of task design and predictions. a.** Incidental encoding task where participants made judgments about whether scenes were indoors (in) or outdoors (out). **b.** Practice betting task where participants made choices based on explicit risk indicated by a color swatch. **c.** Memory-based betting task, where risk level had to be inferred from the position of the stimulus during incidental encoding. **d.** Familiarity judgment phase where participants rated the subjective familiarity of each image on a continuous scale from 1-5, **e.** Source judgment phase, where participants selected the previous quadrant in which the image appeared. **f.** Source confidence judgment where participants rated their confidence on a continuous scale from 1-5. Changes in betting frequency predicted by our hypotheses based on **g.** source accuracy, source confidence and retrieved probability of reward (RPR), and **h.** familiarity, source confidence and RPR. **g.** shows positive main effects of source accuracy, source confidence and RPR, and a three-way interaction between all three terms where betting increases when participants are more confident in correctly recalling that the items were associated with a high reward probability source. **h.** shows positive main effects of familiarity, source confidence and RPR, as well as a two-way interaction between familiarity and source confidence, where the impact of familiarity is diminished when participants are more confident in their recollection of the source at encoding. The example scene image shown here was sourced from the following: https://commons.wikimedia.org/wiki/File:Damavand_in_winter.jpg.

right side from trial-to-trial. Each trial was presented for 4 s. After a decision was made, the text prompt for the selected response was underlined until the end of this period.

## Memory-based betting task

Next, participants were asked to complete a risky-decision task where the values of options depended on information that had to be retrieved from memory (Fig 1c). This task consisted of 338 trials, the first 16 of which were a practice block which was not included in data analysis. The remaining trials were divided into seven blocks of 46 trials. Each trial consisted of a decision phase and two memory judgment phases. In the decision phase, the display was similar to that of the practice risk-based betting task, except a scene image was presented in the center of the screen in place of a color swatch. Critically, the probability of the bet being rewarded depended on the quadrant in which that scene image had appeared during the encoding phase. The association between the different risk-levels and the quadrants was maintained from the practice betting task. The mapping between quadrant and risk was provided by the reference grid appearing at the top of the screen on every trial – as in the practice betting task.

Participants chose to bet on a chance of earning an extra $5 bonus, or to pass and accept a safe $2 bonus, as in the practice betting task. Participants selected a response using their mouse to click on prompts that were randomly positioned on the left or right side of the screen. Participants were asked to use a mouse in this task (unlike the practice betting task) so that they would not need to transfer between the keyboard and mouse in each trial between the decision phase and subsequent memory judgments (see below). The instructions stated that a single trial would be selected at random and their decision to bet or pass would be played out, so each choice was independently incentivized. Participants had 10 s to respond on each trial. After responding, their choice would be highlighted, and the trial would terminate after 1 s.

Following this decision phase, participants were asked to use their mouse to rate their familiarity for each image on a continuous scale from 1–5, with labels of 'sure new,' 'unsure' and 'sure old' at the 1, 3 and 5 positions, respectively (Fig 1d). The image from the betting phase appeared above the scale with a prompt asking, 'How sure are you that this is an OLD or NEW item?' Participants were given 10 s to respond to these familiarity ratings.

If participants indicated that they believed that the item was 'old' (i.e., a familiarity rating greater than 3), then participants were given 10 s to respond to an additional query about the source of the image (Fig 1e). Participants were presented with the image to the left of a small gray 2 x 2 grid representing the same grid from the encoding task. They were asked to click the location in which the image had previously appeared during encoding. Participants were then asked to rate their confidence in this source judgment on a continuous scale from 1–5, with labels 'not at all,' 'somewhat' and 'very sure' at the 1, 3 and 5 positions (Fig 1f).

In addition to the scenes from the encoding task, participants also saw lure images during the memory-based betting task (i.e., scenes that had never appeared during encoding). Because of a coding error, the proportion of lure trials varied slightly for each participant (range 32.9–36.6%, mean = 34.6%, SD = 0.8%). This variance in the proportion of lures was randomized across participants and not systematically linked to any other change in the task conditions. All trials were presented in a randomized order with respect to their lure status, number of presentations and position of presentation in the encoding phase.

We made specific predictions for how participants' task betting behavior would change with item familiarity and source memory, motivated by the hypotheses described above (Fig 1g-h). Namely, we predicted that participants would bet most on items where they could correctly and confidently recall the source of an item, particularly when then value of that source was high, and that participants would bet more on items that they were more familiar with, but this effect would be reduced when they were more confident in the source of the item.

## Participants

Ninety-three participants were recruited via Amazon Mechanical Turk (MTurk) and completed the task on the online experiment platform Pavlovia. Participants were required to have a 90% approval rate on MTurk and had to correctly complete a

simple math question to begin the task. Data was excluded from participants according to preregistered criteria: where accuracy was below 75% in the encoding task (N = 37), or because betting rates were extreme (i.e., greater than 99%, or less than 1%, N = 4). Data from eight participants was lost due to an error in saving data to the Pavlovia server. The average age for the remaining participants (N = 44, 34 male, 10 female) was 38.1 years, SD = 10.7 years. The sample size for this experiment was based on in-laboratory pilot data. In that pilot study, participants completed a task very similar to that described here, with the main differences being that there were no lure items, and participants did not provide familiarity ratings, or make explicit judgments about the source of presented items. Instead, we simply focused on the effect of repeated presentations at encoding on subsequent betting rates. In this pilot, we found a simple effect of item repetition during encoding on betting rates, where the expected effect size was adjusted for an expectation of increased variance in an online sample (estimated Cohen's $d = 0.44$, alpha = 0.05, power = 80%). Participants gave their informed consent to participate in this study via a checkbox, as approved by the Human Research Protections Office at Brown University (IRB#: 0802992441) and were compensated for their participation. The recruitment period for this experiment began on August 20th, 2020 and ended on June 1st, 2022.

## Statistical analysis

Generalized linear mixed effects models (LMEs) with a logit link function were used to estimate the contributions of different memory signals to bet or pass decisions in Matlab 2021a (Mathworks, Natick, MA, USA). We first fit a model to data from all trials with the following formula (LME 1.1, in Wilkinson notation):

$$choice \sim 1 + FM \times lure + participant)$$

Where *FM* corresponds to the familiarity ratings standardized within each participant, *lure* corresponds to a categorical variable coding whether an item was a target or lure (0 or 1). This model also included all main effects for the included terms. This model thus allowed us to test whether betting rates were influenced by familiarity, and if this influence of familiarity depended on whether or not an item was a lure.

To test if betting rates were related to false familiarity on lures specifically, we also modeled the influence of familiarity on choice behavior on lure trials separately using a model with the formula (LME 1.2):

$$choice \sim 1 + FM + participant)$$

We examined the relationship between source memory, familiarity and decision-making for target items that were deemed 'old' by the participants in a separate LME with the following formula (LME 1.3):

$$choice \sim 1 + FM \times RPR \times SC + RPR \times SC \times SA + participant)$$

Where *RPR* corresponds to the retrieved probability of reward based on the chosen source, *SC* corresponds to source confidence and *SA* corresponds to source accuracy. All two-way interaction and main effects were also included.

The complete results of these LME models are provided in S2–4 Tables.

## Preregistration, data and code availability

The study hypotheses and experimental design were pre-registered prior to the start of data collection (https://osf.io/6qfdh). Pilot data was collected for this project prior to preregistration, but is not included in any analysis here.

## Results: Experiment 1

We first confirmed the effect of the repetition manipulation on memory measures, summarized in Table 1. Participants' familiarity ratings and source confidence ratings significantly increased with the number of item presentations. Source accuracy also numerically increased, though this change only reached a trend level for statistical significance.

We estimated participants' retrieved probability of reward (RPR) based on the source they chose for items that they had deemed as being familiar. The RPR of items also increased with the number of presentations, indicating a tendency for participants to attribute higher subjective value to items that had been presented more frequently. However, the mean RPR was also lower than expected from a uniform distribution of choices across the four sources. Indeed, the mean RPR was significantly below the mean probability of reward across sources across all presentation conditions (one sample t-tests: t's ≥ 8.24, P's ≤ 0.0001, Bonferroni corrected, Cohen's d's ≥ 1.24). Participants were thus systematically biased in reporting that items were associated with the lower reward probability sources, and to reporting relatively higher value sources for items that were presented more often.

Absent any information from memory, a rational actor would calculate the probability of an option being rewarded as the average of the probabilities associated with each source (i.e., 50%). Thus, the expected value of betting would be identical for each trial ($2.50). As the safe option had a lower expected value ($2.00), it would be rational to bet on every trial. However, given the known risk-averse preferences of human agents in such settings [28, 29], we anticipated that participants would bet less frequently than would be rational according to this strategy.

To this end, we examined the overall risk tolerance across all trials in the memory-based betting task and practice betting task by examining the intercept term in a model fit to all trials (LME 1.,1). There was a significant negative effect for the intercept term (OR = 0.38, 95% CI: 0.20–0.71, P = 0.003), indicating a general tendency to avoid betting in the memory-based betting task (mean betting rate = 0.42, SD = 0.24), despite the overall higher expected value of betting independent of source.

Familiarity could influence value judgments in this task in at least two ways: it could differentiate target items from lures, and thus provide a clue that an item is associated with some source and hence have some probability of being rewarded. Familiarity could also confer increased value to an item through direct integration into an expected value signal. As predicted by our first hypothesis, there was a strong relationship between betting rates and familiarity (Fig 2a), with participants tending to bet more on items that they had rated as highly familiar (OR = 2.93, 95% CI: 2.04–4.19, P < 0.001). Participants also bet less on lure items compared to targets overall (OR = 0.78, 95% CI: 0.66–0.92, P = 0.003). Nonetheless, the relationship between betting rates and familiarity was maintained for lure items when tested on their own (OR = 1.50, 95% CI: 1.08–2.08, P = 0.01), and there was no interaction between target/lure status and familiarity. (OR = 0.81, 95% CI: 0.57–1.13, P = 0.2). Thus, participants' betting rates increased along with perceived familiarity, even in absence of exposure (Fig 2b).

Unlike mere familiarity, the source of an item was informative about the expected value of an option. We anticipated that participants would weigh the reward probability of the retrieved source in their decision, conditioned on their confidence in their source memory. To examine the role of source memory during choice, we specifically focused on target items that participants had correctly judged as 'old,' and then were queried about their source. Here we could examine how betting was adjusted according to participants' RPR, their confidence in source judgments, and the objective accuracy of their source choice. As expected, participants bet more on items when they believed they had appeared in a source location associated with a higher probability of reward (OR = 2.13, 95% CI: 1.25–3.64, P = 0.008). Participants also

**Table 1. Familiarity and source memory measures across presentations.**

| Measure | Lure (M, SD) | One presentation (M, SD) | Three presentations (M, SD) | Statistics |
|---|---|---|---|---|
| Familiarity ratings | 2.59 (0.93)* | 3.11 (0.72) * | 3.61 (0.65) * | $F(2,86) = 60.59$, $P < 0.0001$, $\eta^2 = 0.58$ |
| Source Confidence | 2.93 (1.13)* | 3.22 (1.01)* | 3.45 (0.85)* | $F(2,86) = 31.05$, $P < 0.0001$, $\eta^2 = 0.42$ |
| Retrieved probability of reward (RPR) | 0.16 (0.18)* | 0.24 (0.15)* | 0.32 (0.14)* | $F(2,86) = 47.40$, $P < 0.0001$, $\eta^2 = 0.52$ |
| Source Accuracy | – | 0.29 (0.09) | 0.32 (0.14) | $t(43) = 1.94$, $P = 0.06$, $d = 0.30$ |

*P < 0.001, Bonferroni corrected post-hoc repeated measures t-test against all other levels

bet more on items when they reported greater source confidence (OR = 1.52, 95% CI: 1.24–1.85, P < 0.0001). The two-way interaction between source confidence and RPR was also significant, though only at the margins of our threshold (Fig 2c; OR = 1.20, 95% CI: 1.00–1.44, P = 0.05). Thus, participants bet more on items when they retrieved a higher probability of reward and were more confident in this information.

We also examined the interaction of RPR, source confidence with source accuracy to test if this relationship changed on trials where participants recalled the correct source. There was also no main effect of overall source accuracy (OR = 0.99, 95% CI: 0.77–1.28, P = 0.6), nor any significant interaction between source accuracy, RPR and source confidence (P's ≥ 0.1).

We had hypothesized that when participants retrieve source information about the expected value of an option, the influence of other less specific signals like mere familiarity should be dampened. We thus tested if the contribution of familiarity to betting choices changed as a function of recollection in this same analysis of target items with reported high familiarity. In contrast to our hypothesis, if anything there was a trend toward a facilitative effect of familiarity and source confidence that did not reach significance (Fig 2d; OR = 1.11, 95% CI: 1.00–1.24, P = 0.06). There was no significant

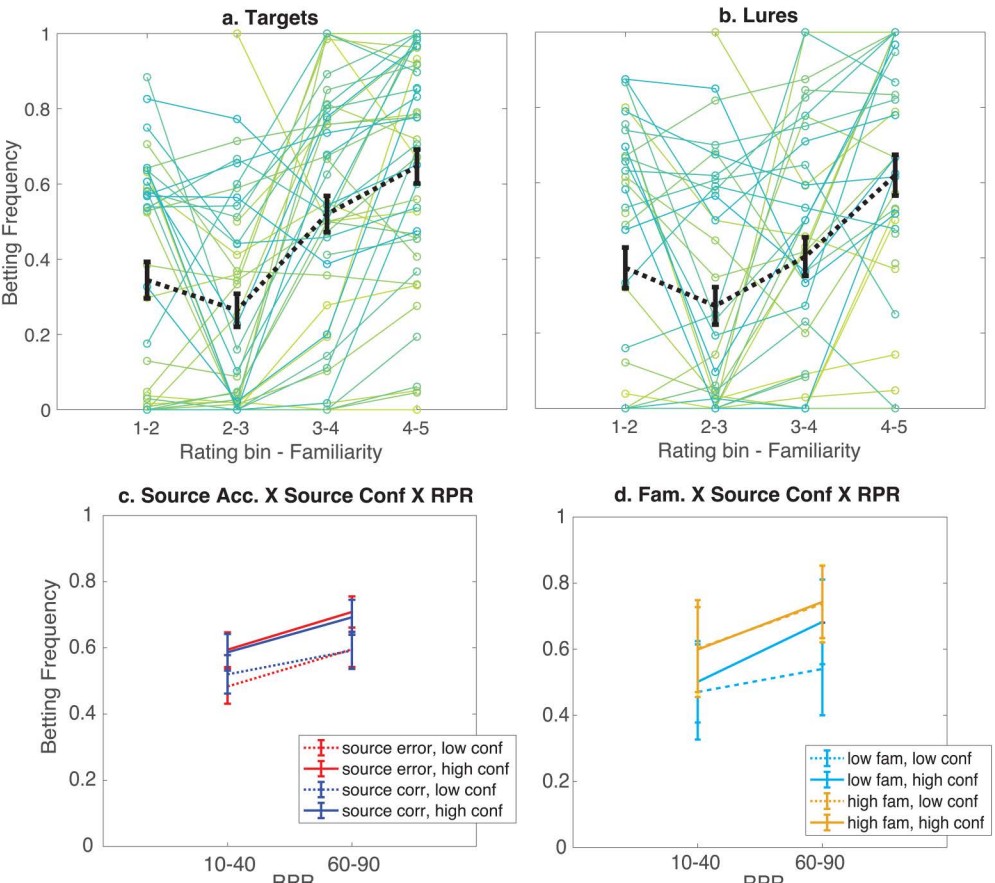

**Fig 2. Behavioral results from Experiment 1.** Panels a and b show betting frequencies as a function of item familiarity ratings in Experiment 1. **a.** Target items, **b.** Lure items. Familiarity ratings have been binned for easier visualization. Individual solid green and blue lines represent means for individual participants. Dashed black line represents the mean across participants. **c.** Mean betting frequency as a function of source accuracy, source confidence (low: source confidence rating ≤ 3, high: rating > 3) and retrieved probability of reward (RPR; 10% or 40% (10–40), 60 or 90% (60–90)). **d.** Mean betting frequencies for target items as a function of familiarity (low: familiarity ratings ≤ 3, high: ratings > 3), source confidence and RPR. Error bars in all panels represent the standard error of the mean.

interaction between familiarity and RPR, nor any three-way interaction between familiarity, RPR and source confidence (P's > 0.3).

## Discussion – Experiment 1

In this first experiment, we confirmed our hypothesis that subjective familiarity increases participants' tendency to bet. These results replicate the pattern observed in our previous study [23], wherein participants were generally averse to betting on items that had not been observed before or had not been encountered for a long time. Our results also extend the classic mere exposure effect [18,19], demonstrating that the subjective feeling of familiarity increases the value of lure items to which participants were never exposed on the encoding phase.

As expected, participants also used recalled source information during decision-making – choosing to bet on options that they believed had appeared in sources associated with a higher probability of reward. As hypothesized, participants weighed this information according to their internal confidence, and were more willing to bet on items when they expressed greater source confidence overall. These results indicate that participants likely integrate meta-memory signals into their value expectations, whether these come from memory of the source or mere familiarity. Thus, their decisions are not simply based on remembering relevant details but on their evaluation of their memory as a source of evidence.

The interactions between familiarity and source memory were different than we had hypothesized. Contrary to our expectations, the effect of familiarity was not reduced for successful recollection of the relevant source, or by confidence in retrieved source information. If anything, there was a trend towards an over-additive interaction between source confidence and familiarity. Thus, rather than discount item familiarity when they were more confident about the source, participants appear to be more willing to bet when both ratings were congruent. This relationship could potentially arise from both ratings reflecting a common, generic memory strength signal [30], or from interactions between familiarity and the process of source retrieval.

Familiarity and source memory also appeared to become linked through increased value associations for more familiar items. Items with more repetitions were attributed to higher value sources. One possibility for this pattern of behavior is that participants' source judgments reflect an increase in the value associated with these items in memory which influences post-retrieval evaluative processes. This pattern is similar to that of past studies that have shown that participants attribute greater value to items that they can retrieve from memory [14,15]. Another possibility is that these source judgments were confirmatory and reflected the betting behavior of the participants.

Importantly, however, limitations in the design of this first experiment complicate strong conclusions. In particular, the memory judgments were made immediately after participants made a bet or pass decision. This close temporal proximity could increase the chance that participants' memory judgments would be influenced by their choice (rather than the opposite), complicating interpretation of any causal role for these memory signals during choice. Second, by thresholding source memory judgments using familiarity, we reduced the amount of data available to test the effects of source memory. This choice may have also limited variance along dimensions like source accuracy and source confidence, possibly explaining why betting rates did not vary with these measures, and why we did not observe an expected increase in source accuracy with repetitions. This choice to threshold source memory judgements by familiarity may have also encouraged a strategy by which weaker items were rated as even less familiar to avoid a cognitively demanding source decision task. This design also limited the range of familiarity signals included in these analyses to items that were perceived as 'old,' which hampered our ability to test the interaction of familiarity and source memory on decision-making. These limitations similarly impacted our ability to test the role of false source memories for lure items.

## Experiment 2

In Experiment 2, we aimed to replicate and extend the results of the first experiment by making changes to the task that would address the limitations described above. First, memory judgments and betting decisions were presented in two

separate tasks, reducing the potential for participants to make memory judgments that conform to their decisions. Second, the order of memory judgments and betting decisions were counterbalanced across participants, so that we might control for any order effects for these two tasks and potential confirmatory biases for making choices consistent with rating and vice-versa. Third, participants made familiarity and source memory judgments in two separate phases of the experiment, and queries about the source were made about all items, old and new, regardless of their rated familiarity. This allowed us to break the interdependency between familiarity ratings and source judgments. Lastly, Experiment 2 was tested first online and then in an in-laboratory sample to address concerns about the quality of data in the online sample (described in further detail below).

## Methods: Experiment 2

### Participants

Two samples were recruited for this experiment through separate mechanisms: an online sample recruited through MTurk and an in-laboratory sample that was recruited from the local community at Brown University. For the online sample, one hundred and seventy-three participants completed the task on the online experiment platform Pavlovia. Requirements for participation were made stricter than the earlier online experiment with the aim of ensuring that more participants would adhere to task instructions: Participants were required to have a 95% approval rate and have completed at least 50 HITs on MTurk in addition to completing a simple math experiment. In keeping with our preregistered methods, data was excluded for participants where accuracy was below 75% in the encoding task or a substantial percentage of responses were missing (N = 59), or because betting rates were highly extreme (betting or passing on greater than 99% of trials, N = 9). Data from 17 participants was lost due to an error in saving data to the Pavlovia server. The remaining 88 participants (59 male, 28 female and 1 participant who did not disclose their gender) had a mean age of 30.0 years (SD = 3.2 years). Participants gave their informed consent to participate in this study via a checkbox, as approved by the Human Research Protections Office at Brown University and were compensated for their participation. The recruitment period for this experiment began on October 14th, 2020 and ended on December 21st, 2021.

The behavior of participants in the online sample indicated a failure of most participants to follow task instructions, particularly in engaging with the more demanding recollection of source memories during decision-making (see below). We thus also collected data from an in-laboratory sample of participants that we analyzed in the same way. Thirty-six participants were recruited for this in-laboratory replication of the online study (24 female, 12 male), with a mean age of 19.2 years (SD = 1.8 years). The sample size for this in-laboratory sample was based on the repetition effect on betting rate observed in the same in-laboratory pilot data used to determine the sample size for the online experiments (estimated Cohen's $d = 0.99$, alpha = 0.05, power = 80%). All participants in the in-laboratory sample had normal or corrected-to-normal vision and were screened for colorblindness, had no current psychiatric or neurological illness, and were not currently taking any psychoactive medications. Participants gave their written informed consent to participate in this study, as approved by the Human Research Protections Office at Brown University and were compensated for their participation. The rate of compensation was the same for the in-laboratory and online samples. The recruitment period for this experiment began on March 24th, 2021 and ended on March 6th, 2022.

### Experimental design

The experimental design was very similar to that of Experiment 1, with a few important changes. As in Experiment 1, all participants first performed an incidental encoding task and practice betting task to familiarize them with the process of making bet or pass decisions based on reward probability. Participants were assigned to either a 'Memory-Betting' group where they were first asked to make familiarity and source judgments on all target and lure items before proceeding to the betting phase, or a 'Betting-Memory' group where the betting task preceded these memory judgments. This assignment

was counterbalanced so that participants were evenly split between these two groups in both the in-laboratory (N = 18 per group) and online samples (N = 44 per group).

Unlike in Experiment 1, the familiarity and source judgments were made in separate phases, each consisting of two blocks of 184 trials. Participants made source memory judgments on all target and lure items, independent of their familiarity ratings. However, the familiarity rating task always preceded the source judgment task for both groups. The format of these tasks was identical to the source and familiarity rating probes in Experiment 1. However, for the in-laboratory group, the position of the mouse was reset to a location in the center of the screen beneath the Likert scale and source options at the start of each decision to minimize the degree to which the mouse position from the previous trial from influenced behavior on the current trial. Controlling the position of the mouse was not possible for the online sample.

The value of the safe option was also reduced to $1.50 from $2 in Experiment 1, as some participants in the previous experiment displayed very low overall betting rates across all trials – making it difficult to interpret how the memory conditions impacted their evaluation of each choice. We thus decreased the value of the safe option to encourage participants to consider the risky option more and engage further with the recollective processes needed to assess its value.

As with Experiment 1, the percentage of lure and target trials varied somewhat for each participant due to a programming error. Variance in the percentage of lures was randomly distributed across participants and not linked to changes in the proportion of other conditions. These percentages did not substantially differ between within the online or in-laboratory samples in the 'Betting-Memory (online: range: 32.3–36.3%, mean = 34.6%, SD = 0.9%; in-laboratory: range: 33.8–37.0%, mean = 35.2%, SD = 0.9%) or 'Memory-Betting (online: range: 32.5–37.0%, mean = 34.7%, SD = 1.1%; range: 33.5–37.3, mean = 35.1%, SD = 0.9%) groups.

## Statistical analysis

As in Experiment 1, LMEs with a logit link function were fit to participants' betting data in both samples. LME 2.1 included all trials and modeled the main effects of familiarity, source confidence and RPR, as well as their interaction with lure status and task ordering, with the following formula:

$$choice \sim 1 + FM \times lure + RPR \times lure + SC \times lure + FM \times order + RPR \times order$$
$$+ SC \times order + lure \times order + (1 + FM \times lure + RPR \times lure$$
$$+ SC \times lure| participant : order)$$

Where *order* refers to a categorical variable specifying if participants were in the 'Betting-Memory' or 'Memory-Betting' group. This model also included all main effects of the terms included in these two-way interactions.

We also fit separate models to target and lure items. For target items (LME 2.2), we tested interactions between familiarity, source confidence and RPR, as well as those between source confidence, source accuracy and RPR using a model with the following formula:

$$choice \sim 1 + FM \times SC \times RPR + SC \times SA \times RPR + order \times FM + order \times RPR$$
$$+ order \times SC + order \times SA$$
$$+ (1 + FM \times SC \times RPR + SC \times SA \times RPR|participant : order)$$

This model included all two-way interactions and main effects within the specified higher order interactions.

We tested a similar model for lure items (LME 2.3), though without the source accuracy factor. As in the previous experiment, this model allowed us to test for relationships between these different factors and betting behavior specifically on lure trials alone. This model had the following formula:

$$choice \sim 1 + FM \times SC \times RPR + order \times FM + order \times RPR + order \times SC$$
$$+ (1 + FM \times SC \times RPR \mid participant : order)$$

We also conducted exploratory analyses of these effects in each task order group using a model with the following formula (LME 2.4):

$$choice \sim 1 + FM \times SC \times RPR + participant)$$

All model formulae included two-way interactions and main effects implied by the specified higher order interactions. The complete results of these LME models for both the online and in-laboratory cohorts are provided in S5–14 Tables.

We also computed participant-wise metrics of familiarity performance from a model of familiarity based on signal detection theory. Specifically, we calculated d-prime metrics for each participant by taking the difference between z-scores corresponding to the hit and false alarm rates expected from the cumulative frequencies of these responses based on a standard normal distribution.

We also conducted secondary analyses examining the relationship of familiarity and source confidence. Exponential and linear functions were fit to the relationships between familiarity and source confidence ratings. In both cases, the functions were fit using the Matlab function 'fminunc' with a cost function based on the negative log likelihood estimated using a Gaussian probability density function. To compare the linear and exponential functions, we carried out a 2-fold cross-validation procedure where we randomly assigned half of the trials to either training and test sets, fit the function in the training set and then estimated the negative log-likelihood for the estimated parameters in the test set. We repeated this procedure 100 times for each participant so that different combinations of trials could be used in either training to test sets. To avoid local minima, the models were fit 10 times for each participant with the parameters initialized at starting points randomly sampled from independent standard normal distributions.

### Preregistration, data and code availability

The study hypotheses and experimental design were pre-registered prior to the start of data collection (https://osf.io/wqfh2).

### Results: Experiment 2

As mentioned above, the behavior of participants in the online sample indicated that a sizeable percentage of participants were not compliant with the task instructions. In particular, betting rates did not vary significantly as a function of the retrieved probability of reward, suggesting that they were not engaging with recalling the source of each item during the memory-based betting phase of this new version of the experiment where memory judgments and value-based choices were separated. Further, the overall performance of this group on the familiarity and source memory tasks was poor. Given that we had little evidence that online participants followed the task instructions, results from the online sample are hard to interpret regarding the experiment's goals. Nevertheless, we describe the results of the online study in the Supporting Information to provide a more comprehensive accounting of our results and avoid contributing to the 'file drawer problem' [31].

We reasoned that an in-laboratory sample would be more likely to comply with the task instructions, and thus reran the same study design in-laboratory with the same analyses. Below we describe the results from the in-laboratory sample.

As with the original experiment, we first confirmed the effect of our repetition manipulation on memory measures (Table 2). Familiarity, source confidence and source accuracy all increased with presentations, confirming the effectiveness of this manipulation in improving memory quality overall.

Unlike Experiment 1, the RPR did not increase over repetitions and the mean RPR across participants was not significantly different from chance in any presentation condition ($P$'s ≥ 0.2). Thus, with the separation of the betting and rating phases, participants did not attribute greater value associations to items that had been more strongly encoded and did not demonstrate any systematic bias in the sources or values with which they associated these items.

Next, we will describe the results of an LME model (2.1) fit to all choices that included main effects for familiarity, source confidence and RPR. With this model we tested whether these effects changed as a function of lure status or task order (i.e., whether participants completed the betting phase or memory phase first). As in Experiment 1, we tested overall risk tolerance in the memory-based decision task by examining the intercept term of this model. There was a significant effect of the intercept term, indicating that participants tended to avoid betting overall (OR = 0.40, 95% CI: 0.25–0.64, P = 0.0001), similar to Experiment 1.

Participants bet much more on items that they rated as familiar (Fig 3a; OR = 1.93, 95% CI: 1.50–2.47, P < 0.0001), and less on lure items (OR = 0.42, 95% CI: 0.29–0.60, P < 0.0001). However, as with Experiment 1, the interaction of lure status and familiarity was not significant (Fig 3b; OR = 0.92, 95% CI: 0.74–1.15, P = 0.5).

Participants also bet more on items with a higher RPR (OR = 1.38, 95% CI: 1.19–1.60, P = 0.0001), and on items for which they had higher source confidence (OR = 1.36, 95% CI: 1.18–1.60, P < 0.0001). The relationship between RPR on betting was decreased for lure items compared to targets (OR = 0.87, 95% CI: 0.76–0.99, P = 0.04), though the influence of source confidence was not significantly changed for lures (OR = 0.90, 95% CI: 0.77–1.05, P = 0.2). There was no difference in overall betting rates for the 'Betting-Memory' and 'Memory-Betting' orderings of the task (OR = 1.61, 95% CI: 0.83–3.10, P = 0.1). Participants tended to bet more on lure items when completing the 'Memory-Betting' task order, but this effect did not reach statistical significance (OR = 1.62, 95% CI: 0.98–2.71, P = 0.06). The effects of familiarity, source confidence and RPR also did not differ as a function of task order ($P$'s ≥ 0.6).

Next, we will describe the results of a second LME (2.2) model that focused on target items, testing interactions between familiarity, source confidence and RPR, as well as source confidence, RPR and source accuracy. There was no overall main effect for source accuracy (OR = 1.09, 95% CI: 0.91–1.23, P = 0.4), nor any significant two-way interaction between source accuracy and source confidence on betting rates (OR = 0.95, 95% CI: 0.83–1.10, P = 0.5). However, there was an interaction between source accuracy and RPR such that participants in the in-laboratory sample bet more when they were correct about the item belonging to a source with a high probability of reward (Fig 3c; OR = 1.18, 95% CI: 1.04–1.34, P = 0.01). This effect was also modulated by their confidence in this source decision, indicated by a significant three-way interaction of source accuracy, RPR, and source confidence (OR = 1.27, 95% CI: 1.10–1.46, P = 0.0009). Participants thus appeared to use information from source memory during the decision task and weighed this recalled information based on their own confidence and the accuracy of recall.

We next tested our hypothesis that the influence of familiarity on choice should decrease when participants have access to more specific and relevant information about expected value retrieved from source memory. Again, in contrast to our expectations, there was no evidence of this negative interaction. Moreover, in this experiment, there was a strong,

**Table 2. Familiarity and source memory measures across presentations.**

| Measure | Lure (M, SD) | One presentation (M, SD) | Three presentations (M, SD) | Statistics |
|---|---|---|---|---|
| Familiarity ratings | 2.28 (0.64)* | 3.32 (0.50) * | 4.19 (0.51) * | $F_{(2,70)}$ = 235.41, P < 0.0001, $\eta^2$ = 0.75 |
| Source Confidence | 2.01 (0.69)* | 2.38 (0.73)* | 2.81 (0.76)* | $F_{(2,70)}$ = 103.03, P < 0.0001, $\eta^2$ = 0.74 |
| Retrieved probability of reward (RPR) | 0.51 (0.09) | 0.51 (0.08) | 0.51 (0.08) | $F_{(2,70)}$ = 0.47, P = 0.6, $\eta^2$ = 0.01 |
| Source Accuracy | – | 0.28 (0.05) | 0.35 (0.09)*** | $t_{(43)}$ = 5.27, P < 0.0001, d = 0.89 |

*P < 0.001, Bonferroni corrected post-hoc repeated measures t-test against all other levels,

***P < 0.001, within-subject two-tailed t-test.

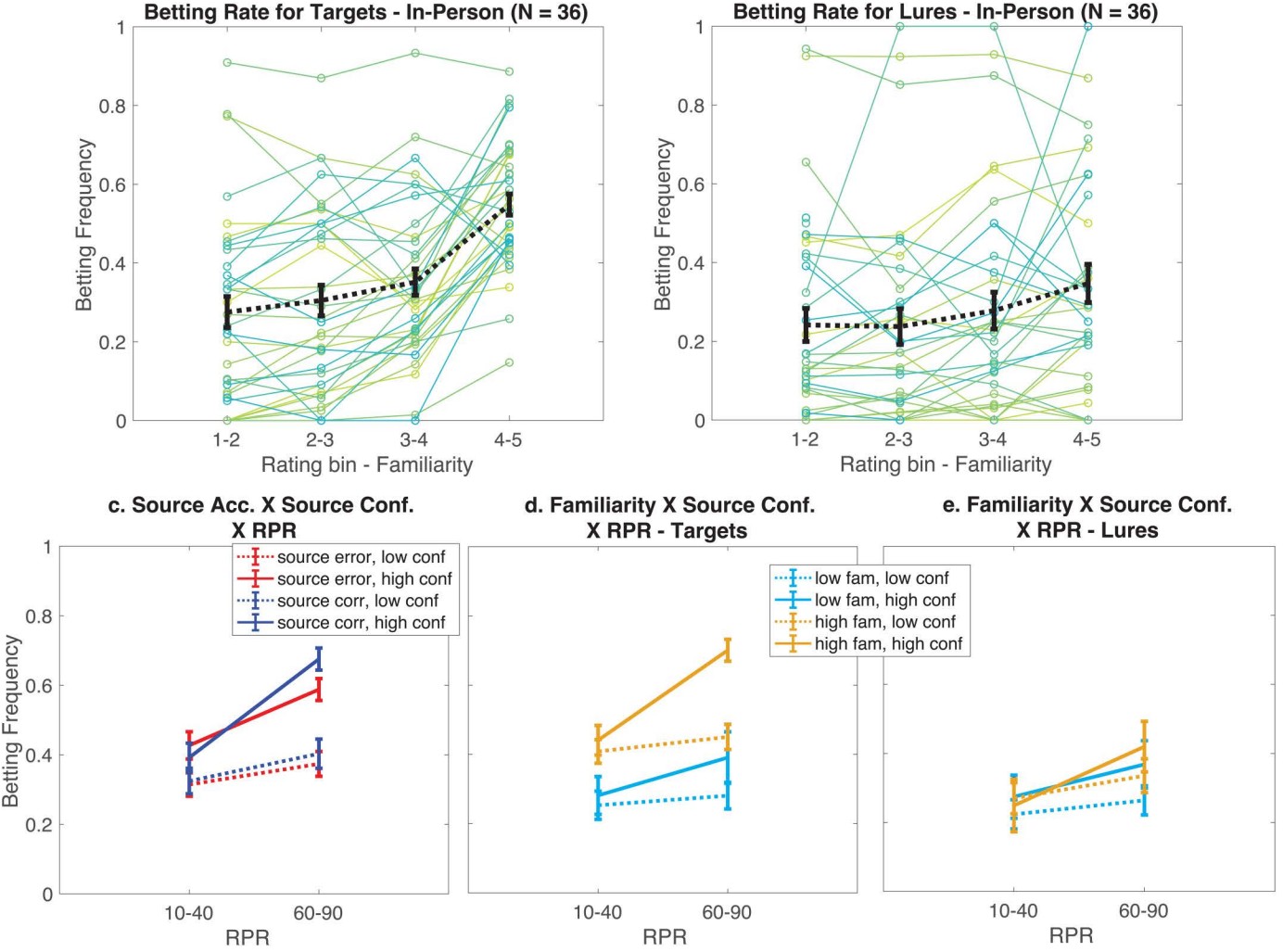

**Fig 3. Behavior from the in-laboratory sample in Experiment 2.** Panels **a.** and **b.** show betting rates as a function of familiarity for targets (**a**) and lures(**b**). Green and blue lines show individual participants, mean and standard error shown in dashed black lines. **c.** Betting rates as a function of the three-way interaction of source accuracy (correct or error), confidence (low: source confidence rating ≤ 3, high: source confidence rating > 3) and retrieved probability of reward (RPR) (10% or 40% (10–40), 60 or 90% (60–90)). Panels d. and e. both show interaction between familiarity ratings (low: rating ≤ 3, high: rating > 3), and source confidence for target items (**d**), and for lures (**e**).

positive, over-additive three-way interaction between familiarity, RPR and source confidence (Fig 3d; OR = 1.12, 95% CI: 1.03–1.21, P = 0.007), such that participants bet most when all three of these factors were higher. That is to say, participants bet much more when they reported an item was familiar, was associated with a high reward source and they were confident in this source. There were also significant positive two-way interactions between RPR and source confidence (OR = 1.11, 95% CI = 1.02–1.21, P = 0.02), as expected, and between familiarity and RPR (1.11, 95% CI: 1.02–1.20, P = 0.01). There was no interaction between familiarity and source confidence (OR = 1.03, 95% CI: 0.94–1.12, P = 0.5). Thus, participant betting increased overadditively when they reported items were familiar and were confident in the item being from a source associated with a high probability of reward.

We also carried out the same analyses on lure items (LME 2.3). This interaction between source memory and familiarity was not as evident for lure items. (Fig 3e). Unlike targets, betting rates on lures did not vary according to a three-way

interaction between familiarity, RPR and source confidence (OR = 1.01, 95% CI: 0.93–1.10, P = 0.8). There were also no significant interactions between familiarity and source confidence and RPR (P's > 0.08). However, there was a significant interaction between RPR and source confidence, as expected, indicating that participants bet more on lures that they reported had been in high reward sources when confident about this memory (OR = 1.13, 95% CI: 1.02–1.26, P = 0.01). Participants also bet on lure items that they rated as more familiar (OR = 1.64, 95% CI: 1.35–1.99, P < 0.0001). This effect was modulated by task order, with the influence of familiarity on betting rates reduced in participants in the 'Memory-Betting' group (OR = 0.72, 95% CI: 0.57–0.92, P = 0.008).

To establish whether this main effect of familiarity held for lures independent of task order, we carried out an exploratory analysis where we separately tested this same LME in the 'Betting-Memory' and 'Memory-Betting' groups (LME 2.4). The familiarity effect was present in participants who completed the 'Betting-Memory' order (OR = 1.67, 95% CI: 1.37–2.07, P < 0.0001), and trended in the same direction but did not reach statistical significance for lures in the 'Memory-Betting' group (OR = 1.17, 95% CI: 0.99–1.39). Thus, familiarity appeared to drive betting most on lure items in absence of prior or proximal familiarity judgments, or any previous exposure.

## Experiment 2 – relationship of familiarity and source confidence

We had initially hypothesized that familiarity and recollection of value relevant information might trade off in their influence on decision-making, as we anticipated that participants would be less sensitive to familiarity when more relevant information from source memory was available. Instead, we found evidence for an overadditive interaction between recalled probability of reward, source confidence and familiarity such that participants' betting rates raised most when all three of these were high. These measures are not process pure: for example, both may reflect a common recollective processes, or source confidence ratings could reflect a general confidence in item memory that is not specific to the studied source [30]. We hypothesized that if both source confidence and familiarity ratings are influenced by a common recollection process, then the relationship between these measures might be stronger on correct source memory judgments of targets compared to source errors and lure items. We further predicted that if recollection has non-linear dynamics (i.e., an 'all or nothing' process), then this relationship might better resemble an exponential function – where the relationship between familiarity and source confidence is flat for lower ratings, but very strong for items with higher ratings.

We first assessed whether the relationship between source memory and familiarity could be better explained by an exponential or linear function. We focused this analysis on correct source judgments of target items, as we anticipated that the recollective process would be most pronounced within a subset of these trials. Using a 2-fold cross-validation procedure, we fit linear or exponential functions to these data and tested them in the other half of the data. Neither model was strongly preferred based on the total negative log-likelihood across participants averaged across fold permutations, though if anything, the linear model provided a slightly better fit (linear = 1275.8, exponential = 1275.9). Comparison of the negative log-likelihood of each model for individual participants also indicated that the linear model was better fit for only a slightly larger percentage of participants (66.7%). The relationship between familiarity and source confidence was thus not obviously better fit by either function, though the linear function was slightly preferred.

Next, we asked whether the relationship between familiarity and source confidence changed as a result of recollection. We compared the slopes for the linear function fit to the familiarity and source confidence ratings for target items where participants made correct and incorrect source judgments, and for lure items (Fig 4). The relationship between familiarity and source confidence had higher slopes for both target items with correct and erroneous source judgments over lures (t(35) ≥ 6.36, P's ≤ 0.0001, Cohen's *d's* ≥ 1.06, corrected), and for items where participants made correct source judgments over errors (t(35) = 5.36, P < 0.0001, Cohen's *d* = 0.89, corrected). The same general pattern of results was observed in the online sample (Supporting Information), and when the slope was fit using an exponential function (not shown).

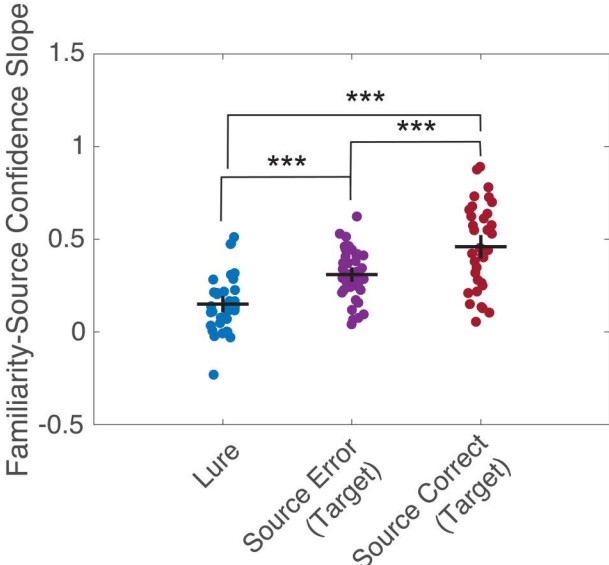

**Fig 4. Slopes for a linear function fit to the relationship between familiarity and source confidence for different item types (lures, targets with correct and incorrect source judgments).** *** $P < 0.0001$, two-tailed within-subjects t-test, Bonferroni corrected for multiple comparisons.

## Discussion – Experiment 2

The second experiment replicated the general effect where participants' betting rates were increased for more familiar target items in both samples. We also extended the findings of the first experiment by testing the interaction between familiarity and value information recalled from source memory across the full range of ratings in both measures, which was not possible in Experiment 1. Participants bet most when they were confident that target items originated from sources associated with a higher probability of reward, and these items were subjectively familiar. This relationship between familiarity and source confidence was stronger for targets compared to lure items in both samples, and generally higher for correct source judgements compared to errors, suggesting that this relationship was strongest when relevant source information was retrieved. However, based on these data alone, we could not reliably determine whether this relationship was linear or exponential. These ratings may both reflect a common recollective process, contributing to a more linear relationship. There may also be individual differences between participants in how much they separate between these constructs in their judgments. Other work has shown that source confidence is not independent of familiarity [30], and source judgments are influenced by confirmatory biases [32]. These factors make it harder to disentangle the underlying processes. However, it appears that participants bet most when they are most confident in an item being veridically associated with a more rewarding source.

Notably, the main effect of familiarity on betting also held for both task order variations, whether betting preceded or followed memory judgements. Thus, we were able to overcome a limitation of the first experiment by showing that the relationship between familiarity and decision-making likely does not just reflect a confirmatory judgement on the part of the participant. Participants also tended to bet more on lures if they made memory judgments about these stimuli prior to the betting task. This behavior could reflect the influence of mere exposure on subjective value [19], or the re-encoding of these novel lures as subjectively old stimuli during the memory judgement phase. However, this effect did not appear to be related to a change in use of familiarity for making betting decisions about these lures. Indeed, if anything, the influence of familiarity on betting for lure items appeared to be reduced for participants who had completed the memory judgment tasks first. Completing these familiarity ratings on lures thus did not create a confirmatory bias for participants

to bet more in line with their previous ratings. Thus, exposure to lures in the memory judgment phase tended to increase participants tendency to bet on these items while also reducing the influence of subjective familiarity for lures. Carrying out these memory judgments might provide a meta-memory benefit by making participants more aware of the presence of these lures, leading them to discount their familiarity during the betting phase. However, this experiment was not designed to have statistical power to compare between the different task order groups, so this interpretation is tentative.

Unlike Experiment 1, we did not observe any effect of repetition on RPR, nor the same overall shift toward more frequent association with lower value sources. These results indicate these effects in Experiment 1 were more likely the result of a confirmatory bias to make source judgments consistent with betting choices than a shift toward associating strongly encoded items with higher value sources. These results strengthen the case that the influence of familiarity on expected value exists independent of changes in memory of item-value associations. These findings highlight the importance of separating choice and memory probes to limit the influence confirmatory biases from influencing participants' judgments.

The absence of an effect of repetition on RPR does not preclude a change in the expected value of items, like the effects observed in other studies [14,15]. Indeed, participants bet more on more familiar items. However, where our study differs from past work is that we did not include value information about items during encoding. Instead, participants had to infer these values from source memory. Hence the information retrieved about items did not include values *per se*, but was instead relevant to calculating an item's expected value.

As noted above, our online sample did not appear to be as engaged with the memory-based aspects of the task. Unlike Experiment 1, participants had to make source and familiarity judgements for every item, unbroken by other tasks – like the memory-based decisions. This made the memory judgments in Experiment 2 longer and more repetitive and may have contributed to online participants disengaging from the task, reducing the quality of their performance.

The differences between participants' behavior in the two samples provides an instructive example of the challenges of running online studies with open-ended problems like value-based decision-making, where multiple behavioral strategies might be considered legitimate. For example, participants in the online sample may have considered the marginal value of carrying out recollection to be lower than that of betting randomly, given the time and cognitive effort that a recollection strategy would require – particularly when further time commitment could come at the cost of not completing other online tasks for additional compensation. We further discuss the interpretation of data from the online sample in the supporting information. While it is typical to increase sample size in anticipation of higher inter-individual variance in an online sample, as we did here, this design provides limited protection against the differences in strategies and priorities for online samples compared to those in-laboratory [Stewart et al., 2017].

## Experiments 1 and 2 - Individual differences analysis

As an exploratory analysis, we tested whether there was any relationship between participants' overall risk tolerance during memory-based betting (measured as mean betting rate) and memory fidelity. We expected that participants who performed worse on the memory tasks would be more risk averse during the memory-based betting task, as the information they were relying on might be more degraded and unreliable. Instead, we found that memory performance (as measured by recognition memory d-prime scores and source accuracy) were inversely correlated with betting rate in Experiment 1, but this pattern was not consistently replicated in either the online or in-laboratory samples (See Supplementary Materials).

## General Discussion

Here we describe two experiments examining the role of different memory processes during risky decision-making. We introduced a novel experimental paradigm that allows for reward probability and strength of encoding to be experimentally orthogonalized and makes it possible to examine the interactions of familiarity and recollection during value judgment.

We had hypothesized that familiarity would increase the perceived value of options, but this effect would be tempered by recollection of relevant source memories. We further hypothesized that recollection and subjective confidence in recalled value information would increase subjective value and increase the precision of decision-making so that participants would make choices more in line with recalled values. We confirmed that participants bet more on familiar items, however contrary to our expectations, the influence of familiarity did not diminish with better source memory. Instead, subjective value for items in memory reflected the overadditive contributions of familiarity, retrieved values and confidence in that retrieval.

Across both experiments, we found that participants' betting rates increased with familiarity across all reward probabilities, in line with the hypothesis that familiarity generally increases subjective value of options, even when unrelated to task goals [18,19]. Notably, we also found betting rates increased with familiarity even in absence of prior exposure (i.e., with lures) in Experiment 1 and in the in-laboratory sample in Experiment 2. This observation across experiments indicates that exposure is sufficient, but not necessary, to increase familiarity-driven changes in subjective value. The familiarity attributed to lures, likely driven by the frequency of their contents in daily life or commonalities with targets [33–35], thus contributes to an increase in value that is not explained by exposure alone. While this effect of familiarity was not observed in the online sample in Experiment 2, the null effect is likely related to the higher behavioral variance in the online sample, and the smaller number of trials for lures relative to targets, providing less samples per participant to sample this intra-individual variance.

Contrary to our initial hypothesis, participants did not rely on familiarity less when they could recall value relevant information from source memory across experiments. One possible explanation for this pattern of results is that familiarity gates the use of source information in making betting decisions – which could explain the overadditive three-way interaction between familiarity, source confidence and retrieved reward probability on betting frequency. Our findings could thus suggest a dual role for familiarity in memory-based decision-making, both in increasing subjective value directly and in mediating the use of retrieved values. This interpretation is in line with evidence that retrieval is gated by familiarity signals, allowing for more selective engagement of this more demanding process when familiarity is higher [36,37]. This function would also be consistent with adaptive memory theories suggesting that retrieval is balanced between the importance of the memory being retrieved and likelihood of retrieval success, and the costs associated with retrieval [16].

An interaction between source memory and familiarity during decision-making might depend on cortico-striatal-thalamic circuitry that has been more broadly implicated in action selection. This circuitry plays an important role in controlling behavior, allowing some actions to go forward and withholding others through a balance of activity in the cortex, basal ganglia and thalamus. Several lines of evidence point toward the striatum playing a key role in this gating function both for motoric and cognitive actions, thus mediating control over behavior [38,39]. Our group has previously proposed the striatum as a hypothesized neurobiological mechanism for gating retrieved information [40]. In particular, across studies, we have previously shown that signals in the striatum are correlated with feedback-related changes in decision threshold during familiarity judgments [41], and with betting decisions for unfamiliar items [23]. This striatal gating mechanism could thus be a common feature of decisions about memory, value and recollection.

An alternative to this gating account is that familiarity and value information in source memory are integrated together into a common scalar value signal. For example, familiarity might have a multiplicative influence on risk, or lower the ambiguity for the range of plausible values, resulting in the same kinds of non-linear dynamics that are well characterized for probability and amount in standard risky decision-making tasks [42,43]. Weilbacher, Kraemer (12) showed that individual differences in memory bias during a decision-making task were not correlated with behavior in descriptive risk and ambiguity tolerance tasks, suggesting that memory biases on decision-making rely on separate processes from those involved in risk and ambiguity tolerance. However, it is possible that ambiguity or risk derived from memory signals could influence decision-making more like experienced risk, where behavior tends to also deviate from that in descriptive risk tasks [44].

Another alternative explanation is that this familiarity signal might increase confidence in decisions based on the value of retrieved information. Decision confidence can increase for both positive and negative values when based on an increased distance from the decision boundary – i.e., when a bet or pass choice is less ambiguous [45]. Similarly, familiarity might increase decision confidence for retrieved values on either side of the decision boundary, sharpening the choice function. However, the average betting rate did not substantially drop for high familiarity, high source confidence decisions with low RPR, arguing against this explanation.

A simpler explanation is that familiarity and source confidence ratings reflect a common recollective process in some trials. Retrieval of source information appeared to influence both familiarity and source confidence ratings, as these measures were correlated, and the strength of this correlation increased in trials where accurate recollection was more likely (correct source judgements), or possible (target items). One explanation for this increased slope is the recollection of highly specific memories from encoding – which could increase both of these measures. Taken together with our findings from the betting phase of the task, participants may bet most when they can recollect that an image was linked to a high reward source.

We also conducted an exploratory analysis examining the relationship between betting rates and memory performance. While we expected that participants with worse memory would bet less frequently, instead we observed the opposite relationship, though this effect was not consistent across experiments and samples. This relationship could arise for several reasons: because these participants were less engaged with the memory components of the task and instead tended toward the more rational strategy of betting more often, or participants who performed worse on these tasks (and perhaps are generally less engaged with the experiment) could also be less risk aversion in general. The fact that these effects are strongest in the online sample in Experiment 1 where the betting and memory component of the task are immediately adjacent also make us suspicious that the correlation between these measures is driven by individual differences in task engagement more than a deeper relationship between memory and these decision processes.

One noteworthy limitation of our experimental design was that we only focused on conditions where there was a potential reward for betting and an opportunity cost for passing. We did not examine any conditions with aversive outcomes or losses where recalled values should actively drive participants away from betting more. For example, Weilbacher, Kraemer [12] showed that participants' bias towards recalled items flips for losses, as they prefer to choose options with an indefinite value than a retrieved loss. However, it is not clear how the familiarity effects observed here would interact with losses. For example, if familiarity is truly serving a dual role, the interaction between familiarity and loss amount might be expected to flip into the negative direction, though the main positive effect of familiarity on betting rates might be preserved.

In both experiments, absent any source memory, the rational course of action is to bet on every trial as the expected value of the risky option is always higher than the safe option. While only a small subset of participants displayed such behavior, they were rejected from data analysis as the lack of variability in their behavior made it impossible to use their data to test questions about the relationship between information in memory and decision behavior. In the future, this task could be modified to include trials with more extreme value independent of source memory so that betting or passing is strictly rational. Such trials would help distinguish between participants who are adopting a heuristic strategy of betting or passing on all trials from those who are making rational economic choices.

Here we examined the separate contributions of familiarity and source memory in a novel decision paradigm that allowed us to test the role of these memory processes on value judgment while removing any potential circular influence of value information on memory strength. Our findings highlight the centrality of familiarity in memory-based valuation and decision-making. Familiarity increases subjective value and appears to positively interact with value information that must be recalled from source memory. Future behavioral or neuroimaging studies may help untangle the mechanisms by which familiarity influences value judgement or examine how the use this information is disrupted by memory impairments. The

paradigm described here provides a promising platform for testing these mechanisms and understanding how dysfunction within memory systems could contribute to decision impairments in neurodegenerative disorders.

## Supporting information

**S1 Appendix. Supporting information.**
(DOCX)

## Acknowledgments

We would like to thank Atsushi Kikumoto and Olga Lositsky for input on the design of this experiment. Additional thanks to Emily Waters for assistance in piloting earlier versions of this task.

## Author contributions

**Conceptualization:** Avinash Rao Vaidya, David Badre.

**Data curation:** Avinash Rao Vaidya, David Badre.

**Formal analysis:** Avinash Rao Vaidya, Johanny Castillo, Alejandro Torres, David Badre.

**Funding acquisition:** David Badre.

**Investigation:** Avinash Rao Vaidya, Johanny Castillo, Alejandro Torres.

**Methodology:** Avinash Rao Vaidya, Johanny Castillo, David Badre.

**Project administration:** Avinash Rao Vaidya, Johanny Castillo, David Badre.

**Resources:** David Badre.

**Software:** Avinash Rao Vaidya.

**Supervision:** David Badre.

**Validation:** Avinash Rao Vaidya.

**Visualization:** Avinash Rao Vaidya.

**Writing – original draft:** Avinash Rao Vaidya.

**Writing – review & editing:** Avinash Rao Vaidya, Johanny Castillo, David Badre.

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
