## [Decision Letter · Decision Letter 0]

30 Dec 2024

Dear Dr. Vaidya,

Thank you for submitting your manuscript to PLOS ONE. After careful consideration, we feel that it has merit but does not fully meet PLOS ONE’s publication criteria as it currently stands. Therefore, we invite you to submit a revised version of the manuscript that addresses the points raised during the review process.

We look forward to receiving your revised manuscript.

Kind regards,

Claudia Greco, Ph.D.

Academic Editor

PLOS ONE

Journal Requirements:

 “DB

Office of Naval Research

MURI-N00014-16-1-2832

https://www.onr.navy.mil/

No

(Award supports first author)

ZIA DA000642

National Institutes on Drug Abuse

https://nida.nih.gov/

No”

4. Please note that your Data Availability Statement is currently missing [the repository name and/or the DOI/accession number of each dataset OR a direct link to access each database]. If your manuscript is accepted for publication, you will be asked to provide these details on a very short timeline. We therefore suggest that you provide this information now, though we will not hold up the peer review process if you are unable.

5. Please ensure that you refer to Figure 4 and 5 in your text as, if accepted, production will need this reference to link the reader to the figure.

6. We note that Figure 2 in your submission contain copyrighted images. All PLOS content is published under the Creative Commons Attribution License (CC BY 4.0), which means that the manuscript, images, and Supporting Information files will be freely available online, and any third party is permitted to access, download, copy, distribute, and use these materials in any way, even commercially, with proper attribution. For more information, see our copyright guidelines: http://journals.plos.org/plosone/s/licenses-and-copyright.

Additional Editor Comments:

Thank you for submitting your manuscript to PLOS ONE. After careful consideration, I feel that it has merit but does not fully meet PLOS ONE’s publication criteria as it currently stands. Both reviewers have recommended minor revisions. While the requested revisions are categorized as minor, I strongly encourage you to address all the reviewers' comments and suggestions in full.

Both reviewers provided positive feedback regarding the overall quality of your study, highlighting the robustness of the experimental design and the relevance of the findings in the field of memory and decision-making. However, they have also raised several points that require further clarification and refinement in the revised version.

Their feedback is insightful and, when implemented, will significantly enhance the clarity, rigor, and overall impact of your manuscript. I believe their comments are highly relevant and aimed at improving the quality of your work, ensuring that it meets the highest standards for publication. Specifically, the reviewers have provided detailed recommendations in the following areas:

The presentation of your experimental design and figures.Justifications for methodological choices, including statistical analyses.Clarifications in your discussion and interpretation of results.

While some of the points raised by the reviewers represent suggestions rather than necessary revisions, I encourage you to consider these recommendations where possible. These suggestions provide an opportunity to enrich your manuscript by introducing interpretations that were not fully explored in the original submission. Doing so will not only address the reviewers' concerns but may also offer new insights and strengthen the overall impact of your study.

I did not identify any major conflicts between the reviewers' comments. Both reviewers raised similar questions regarding statistical modeling and the interpretation of results. My recommendation is to prioritize clarity and provide detailed justifications for your methodological choices. Where applicable, ensure that both reviewers’ concerns are comprehensively addressed in your response letter.

Reviewers' comments:

Reviewer's Responses to Questions

**Comments to the Author**

1. Is the manuscript technically sound, and do the data support the conclusions?

Reviewer #1: Yes

Reviewer #2: Yes

2. Has the statistical analysis been performed appropriately and rigorously?

Reviewer #1: Yes

Reviewer #2: Yes

3. Have the authors made all data underlying the findings in their manuscript fully available?

Reviewer #1: Yes

Reviewer #2: Yes

4. Is the manuscript presented in an intelligible fashion and written in standard English?

Reviewer #1: Yes

Reviewer #2: Yes

Reviewer #1: These two experiments (one fully online and the second both online and in-laboratory) examined different memory processes (i.e., familiarity versus recollection) interact during value judgment and risky decision-making.

The two experiments were well executed, aimed to answer the 4 research hypotheses and the results obtained were promising, and positive which are relevant for implementation of programs. The statistical analyses were appropriate in testing the various hypotheses postulated.

A pilot study was carried out. Perhaps, a brief summary of its findings would be good to illustrate to the readers about this results and how were these incorporated in the actual experiments.

In Experiment !, the analysis showed that there was no main effect of overall source accuracy, nor any significant interaction between source accuracy, RPR and source confidence. What could be the reasons for this? Nor was there any significant interaction between familiarity, RPR, and source confidence.

Same experiment design was use in Experiment 2. How were these participants split into 2 groups (online and in-laboratory groups)?

For the online sample, majority of the participants were not compliant with the task instructions. Why is this so?

Discussion and limitations of study were well discussed for both the experiments.

Reviewer #2: This is a well-written and interesting paper examining the separate and interacting effects of recollection and familiarity on value judgment and memory-based decision making. The authors find that familiarity increases subjective value and that it does not tradeoff with source memory—recollection and familiarity instead interact to both increase subjective value. The authors do a great job motivating the study and explaining their hypotheses, and the study design and analyses are well thought through and technically rigorous. The flow of the paper is also great—for example, I had noted two major weakness of experiment one (the causal interpretation of effects of familiarity on choice and the potential unwillingness of participants to complete the source memory task), but both were stated in the discussion of experiment one and then addressed by experiment two. So I think the paper is in generally good shape, and I have the following comments (which are all mostly minor and given in rough order of when they are relevant in the paper) to help improve it:

- I think it would be more clear to present the prediction figure (Figure 1) after (or alongside) the design of experiment one. This is because the predictions are specific to the experiments, and it would be easier to interpret them once the experiment is explained. I’d like to suggest that the predictions are either added as a separate panel to what is currently Figure 2 (the task design), or that these two figures are reversed in order. I also think that the axes could be improved (the obvious assumption is that left is low and right is high, but this should be made explicit).

- While reading the task design, I kept wondering how the bet structure made sense when it was introduced by the practice betting task (for example, I kept wondering whether losses were involved). This is later clarified in the memory-based betting task, but it’d be good to introduce that earlier. Related to the presentation of the task, Figure 2C is labeled memory-based betting but in the methods that label is used to describe the whole four-part task. I think this should just be made to match, and the authors can solve this however they see fit/what they think is the most clear. The methods also state the task is three-parts, (which I sort of get if both source tasks are grouped together), but I think it should be labeled as having three-partings.

- I have a few clarification questions about the statistical analyses for experiment one. How was the categorical lure predictor coded (0 and 1 or some other coding scheme?) Related to this—why was a separate model for the effects of lure familiarity trials needed (LME 1.2), since these effects should be captured by the interaction term of the previous model (i.e. if the categorical lure predictor is 1)? I am not necessarily concerned by this choice, but I would like it to be clarified. There is also a predictor labeled RSP in LME 1.3, but this was not defined as far as I could tell (and I apologize if I missed this).

- It is interesting that repetitions increased the RPR but not source accuracy (and later that the opposite occurs in experiment 2). I think this may be because these variables are in tension with one another: participants should be less accurate about the source if they tend to mis-attribute low probability sources to high probability sources (which should in turn increase the effect of RPR on betting). Was there any evidence for such a relationship? This is also relevant for the overall discussion (~ lines 676 - 682).

- A nice feature of this design is that it is rational to bet on every trial if participants have no source memory. But isn’t it the case that participants who may be completely rational but not rely on memory are being excluded by the pre-registered criteria (>99% bet rates)? I’m not very concerned by this because very few participants are actually affected here, but I think it might be worth stating why this rationale was chosen given the design of the task. I am also wondering whether you found any relationship between memory performance and overall betting rates. I know you observed that people are generally risk averse here, but I wonder whether it is also the case that less risk averse people are also more forgetful, which would indicate that they are rational when memory is poor? I think this aspect of the design might allow you to gain more insight into participants’ overall strategies when solving the task.

- While the authors admit that drawing causal influences of familiarity on betting behavior is difficult, there are still statements about directionality in the results (lines 275-277; lines 281-282). I think these should be amended. Relatedly, in experiment two the authors state that adding some temporal distance (via separate blocks for decisions and memory judgments) can help address this, but I don’t see how this is necessarily the case. The element of the experiment two design that really accomplishes this is the counterbalancing, so in my opinion the writing should be revised to reflect that as a primary explanation for this design choice in experiment two.

- Regarding the statistical methods for experiment two, I am again unclear why a separate model for lure items is needed, since this should again (I think) be captured by the interactions with the lure categorical predictor variable. I understand why this is the case for targets since the authors need to look at the effects of source accuracy, but the rationale for doing so with lures should be made more clear.

- When comparing the linear and exponential models, a procedure that is more robust to overfitting should be used. Ideally this would be cross validation (for example, leave-N-subjects or, per subject, leave-N-trials out). I think it is possible that this may reveal more nuanced differences in the fit than just a straight comparison of the NLL. This ultimately may not be so important given that the authors state (but don’t show) that a similar result was observed for the relationship between familiarity/source confidence and lures/errors/correct responses, but I think cross validation would strengthen this result.

- A few typos:

- Stars are missing from the source accuracy portion of table 2

- line 235: they value expectations —> their value expectations

- line 525: I think this should be referencing Figure 4a rather than 3a (which was for experiment 1)

- line 528: Same here, I think this should be Figure 4b instead of 3b

- line 547: Figure 3c —> Figure 4c

- line 559: Figure 3d —> Figure 4d

**Do you want your identity to be public for this peer review?** For information about this choice, including consent withdrawal, please see our Privacy Policy

Reviewer #1: No

Reviewer #2: No

---

## [Author Response · Author response to Decision Letter 1]

19 Feb 2025

Please see our response letter for a detailed response to reviewer comments.

---

## [Decision Letter · Decision Letter 1]

25 Mar 2025

Influences of familiarity and recollection on value-based decision-making

PONE-D-24-37092R1

Dear Dr. Vaidya,

We’re pleased to inform you that your manuscript has been judged scientifically suitable for publication and will be formally accepted for publication once it meets all outstanding technical requirements.

Kind regards,

Claudia Greco, Ph.D.

Academic Editor

PLOS ONE

Additional Editor Comments (optional):

Reviewers' comments:

Reviewer's Responses to Questions

**Comments to the Author**

Reviewer #1: All comments have been addressed

Reviewer #2: All comments have been addressed

2. Is the manuscript technically sound, and do the data support the conclusions?

Reviewer #1: Yes

Reviewer #2: Yes

3. Has the statistical analysis been performed appropriately and rigorously?

Reviewer #1: Yes

Reviewer #2: Yes

4. Have the authors made all data underlying the findings in their manuscript fully available?

Reviewer #1: Yes

Reviewer #2: Yes

5. Is the manuscript presented in an intelligible fashion and written in standard English?

Reviewer #1: Yes

Reviewer #2: Yes

Reviewer #1: Comments were addressed. I am happy with the revisions made.

Reviewer #2: The authors have done an excellent job revising the manuscript and carefully considering my comments, and I think it is ready for publication.

**Do you want your identity to be public for this peer review?** For information about this choice, including consent withdrawal, please see our Privacy Policy

Reviewer #1: No

Reviewer #2: No

---

## [Editor Report · Acceptance letter]

PONE-D-24-37092R1

PLOS ONE

Dear Dr. Vaidya,

I'm pleased to inform you that your manuscript has been deemed suitable for publication in PLOS ONE. Congratulations! Your manuscript is now being handed over to our production team.

Kind regards,

on behalf of

Dr. Claudia Greco

Academic Editor

PLOS ONE